# Equity-Specific Effects of Interventions to Promote Physical Activity among Middle-Aged and Older Adults: Development of a Collaborative Equity-Specific Re-Analysis Strategy

**DOI:** 10.3390/ijerph16173195

**Published:** 2019-09-01

**Authors:** Gesa Czwikla, Filip Boen, Derek G. Cook, Johan de Jong, Tess Harris, Lisa K. Hilz, Steve Iliffe, Richard Morris, Saskia Muellmann, Denise A. Peels, Claudia R. Pischke, Benjamin Schüz, Martin Stevens, Frank J. van Lenthe, Julie Vanderlinden, Gabriele Bolte

**Affiliations:** 1Department of Social Epidemiology, Institute of Public Health and Nursing Research, University of Bremen, 28359 Bremen, Germany; 2Health Sciences Bremen, University of Bremen, 28359 Bremen, Germany; 3Department of Movement Sciences, Physical Activity, Sports & Health Research Group, KU Leuven, 3001 Leuven, Belgium; 4Population Health Research Institute, St George’s University of London, London SW17 0RE, UK; 5School of Sports Studies, Hanze University of Applied Sciences, 9747 AS Groningen, The Netherlands; 6Research Department of Primary Care & Population Health, University College London, London NW3 2PF, UK; 7Department of Population Health Sciences, Bristol Medical School, University of Bristol, Bristol BS8 2PS, UK; 8Leibniz-Institute for Prevention Research and Epidemiology—BIPS, 28359 Bremen, Germany; 9Department of Psychology and Educational Sciences, Open University, 6401 DL Heerlen, The Netherlands; 10Institute of Medical Sociology, Centre for Health and Society, Medical Faculty, University of Duesseldorf, 40225 Duesseldorf, Germany; 11Department of Prevention and Health Promotion, Institute of Public Health and Nursing Research, University of Bremen, 28359 Bremen, Germany; 12Department of Orthopedics, University of Groningen, University Medical Center Groningen, 9713 GZ Groningen, The Netherlands; 13Department of Public Health, Erasmus University Medical Center Rotterdam, 3000 CA Rotterdam, The Netherlands

**Keywords:** physical activity, social inequalities, interventions, intervention-generated inequalities, equity impact assessment, middle-aged adults, older adults

## Abstract

Reducing social inequalities in physical activity (PA) has become a priority for public health. However, evidence concerning the impact of interventions on inequalities in PA is scarce. This study aims to develop and test the application of a strategy for re-analyzing equity-specific effects of existing PA intervention studies in middle-aged and older adults, as part of an international interdisciplinary collaboration. This article aims to describe (1) the establishment and characteristics of the collaboration; and (2) the jointly developed equity-specific re-analysis strategy as a first result of the collaboration. To develop the strategy, a collaboration based on a convenience sample of eight published studies of individual-level PA interventions among the general population of adults aged ≥45 years was initiated (UK, *n* = 3; The Netherlands, *n* = 3; Belgium, *n* = 1; Germany, *n* = 1). Researchers from these studies participated in a workshop and subsequent e-mail correspondence. The developed strategy will be used to investigate social inequalities in intervention adherence, dropout, and efficacy. This will allow for a comprehensive assessment of social inequalities within intervention benefits. The application of the strategy within and beyond the collaboration will help to extend the limited evidence regarding the effects of interventions on social inequalities in PA among middle-aged and older adults.

## 1. Introduction

Reducing social inequalities in health has become a priority for public health. Studies have consistently shown social gradients in health, whereby a higher socioeconomic position (SEP) corresponds with better health [1,2]. Social inequalities can be defined as differences between population subgroups represented by socioeconomic and/or sociodemographic characteristics, such as socioeconomic position (SEP), ethnicity, social capital, or gender/sex [3]. These differences have also been found for physical activity (PA) behavior. In this regard, individuals with a low SEP (e.g., low education, occupation, or income), those living in a deprived residential area, ethnic minorities, older individuals without a spouse, as well as females have been found to have lower levels of leisure time PA of moderate or vigorous intensity (e.g., sports, exercising, recreational walking, recreational cycling) [3,4,5,6,7]. Because regular PA has been shown to be an important determinant of health and wellbeing [8], social inequalities in PA have been discussed to play an important role in explaining health inequalities [9].

Interventions to promote PA may be designed to exclusively focus on socially disadvantaged population groups, such as those living in socioeconomically disadvantaged communities [10] or ethnic minorities [11]. These “targeted interventions”, if implemented successfully, may reduce health inequalities by improving PA among the targeted socially disadvantaged population group only [3]. “Universal interventions” targeting the general population rather than specific socially disadvantaged population groups are also considered a promising approach. They have the potential to both benefit large numbers of individuals and help reduce inequalities by benefiting socially disadvantaged population groups disproportionally more [12]. However, evidence suggests that universal interventions, even if successful at improving outcomes across a population, may unintentionally widen health inequalities [13,14,15,16].

Further evidence indicates that universally provided interventions (i.e., not targeted at specific socially disadvantaged population groups) focusing on individual behavior change are more likely to increase inequalities compared to universally provided interventions focusing on changes of social, built, or policy environments [14]. According to the concept of ‘individual agency’ [15], this may be because individual-level behavior change interventions require comparatively more cognitive, psychological, time, and material resources from the individual in order to gain benefit. As these resources tend to be socioeconomically patterned, whereby high-SEP individuals have relatively greater resources than those with a low SEP, interventions are likely to be more beneficial for high-SEP individuals [15,17,18]. Another explanation being discussed is that some of the psychosocial determinants of behavior operate differentially according to SEP [19,20]. Moreover, discrepancies between the perceptions of low-SEP individuals and health promoters regarding health behavior, behavior change, and support for behavior change may also result in interventions being less effective among low-SEP population groups [21]. From a theoretical point of view, the same may hold true for interventions in which different population subgroups are not treated equally, with higher-SEP individuals receiving preferential treatment.

In 2007, Whitehead and colleagues [22] called for all future public health interventions to be analyzed for their impact on health inequalities, including those related to SEP and gender. Since previous studies indicate gender differences in the domains in which people prefer to engage in PA as well as in motivating factors and context preferences for PA [23,24], it seems plausible that interventions may also be differentially effective among males and females. However, in studies of PA interventions, equity-specific intervention effects have rarely been evaluated [25,26,27].

Humphreys and Ogilvie [25] conducted a pilot review on equity-specific effects of environmental and policy interventions to promote PA. They found only a limited number of primary studies reporting differential effect analyses by at least one indicator of social inequalities other than gender. With regard to gender inequalities, intervention effects appeared to be evenly distributed in about half of studies, while the other half pointed to gender-specific intervention effects suggesting that some interventions may affect males and females differently. In a systematic review of randomized controlled trials, Attwood and colleagues [26] investigated differences in the effects of primary-care-based PA interventions across indicators of social disadvantage among adults. Like Humphreys and Ogilvie [25], they also identified only a few studies that reported the details of relevant analyses. As a consequence, firm conclusions concerning the impact on social inequalities in PA could not be drawn [26]. The scarcity of studies reporting equity-specific effect analyses have also been reported in another equity-focused systematic review on interventions to promote PA among adults aged ≥50 years by Lehne and Bolte [27]. They also found that equity-specific analyses, when reported, were primarily oriented towards gender comparisons, with five studies indicating that some interventions may affect males and females differently [27].

All three reviews concluded that studies often collect sufficient information on indicators of social inequalities to permit equity-specific intervention effects to be investigated. However, the majority of studies do not report having analyzed equity-specific intervention effects, indicating that the potential for assessing the impact of interventions on social inequalities in PA has not yet been exploited [25,26,27]. Assessing the impact of interventions on health inequalities requires interaction or subgroup analyses to compare intervention effects across different population subgroups [28]. However, due to limited resources, most studies are not prospectively designed with sufficient power to examine effects in subgroups which may limit credibility of equity-specific findings [28]. Although insufficient statistical power is a valid reason for not to conduct or report such equity-specific analyses, the importance of understanding how interventions affect health inequalities make re-analyzing intervention studies by indicators of social inequalities a potentially valuable approach [29,30,31,32,33]. As this requires access to and analyses of primary data, a collaborative approach involving authors and researchers of the primary studies is necessary.

The prevalence of PA tends to decline with increasing age and is particularly low in midlife and older adults [34,35], who are an important target group for PA promotion. Interventions to promote PA among middle-aged and older adults are increasingly being implemented. However, evidence of their impact on social inequalities in PA is scarce. To avoid unintentionally inducing or increasing existing social inequalities in PA and PA-related health outcomes in the middle-aged and older population, it is necessary to evaluate whether the effects of especially individual-level interventions differ by relevant socioeconomic and sociodemographic characteristics. Therefore, the aim of this study is to develop and test the application of a strategy for re-analyzing equity-specific effects of existing individual-level PA intervention studies in middle-aged and older adults, as part of an international interdisciplinary collaboration. This article aims to describe (1) the establishment and characteristics of the collaboration; as well as (2) the jointly developed equity-specific re-analysis strategy as a first result of the established collaboration.

## 2. Materials and Methods 

### 2.1. Context

This study is carried out as part of the subproject “EQUAL—Equity impacts of interventions to increase physical activity” within the prevention research network “AEQUIPA—Physical activity and health equity: primary prevention for healthy ageing” [36]. The overall aim of AEQUIPA is to strengthen the evidence base for PA and PA promotion in the context of healthy ageing and health equity. Besides the subproject EQUAL, AEQUIPA comprises five further subprojects, one of which is “PROMOTE—Tailoring physical activity interventions to promote healthy ageing”. Within this subproject, the effects of two web-based tailored PA interventions in older adults aged 65–75 years were examined and compared to a delayed intervention control group [37,38].

EQUAL aims to investigate equity-specific intervention effects combining the results of the PROMOTE intervention trial with the results of previously conducted PA interventions using equity-specific re-analyses. Other projects on equity-specific re-analyses either asked responsible researchers of previous studies to provide their study data for a re-analysis or conducted the re-analyses based on the studies’ original analytical strategies [30,33,39]. Going beyond these approaches, EQUAL seeks to cooperate closely with researchers of the studies included in the collaboration and to jointly develop a strategy to be used by the researchers for performing the re-analyses of their own data (i.e., the individual participant data of the collaborating studies will not be pooled), harmonizing data across studies as much as possible. For this purpose, the EQUAL project includes two face-to-face workshops to bring together the participating researchers for discussing comparability of data, jointly developing the analysis strategy, as well as discussing results and deriving implications for future interventions.

### 2.2. Establishment of the Collaboration

#### 2.2.1. Search Strategy

To develop the strategy, we aimed to initiate a collaboration based on a sufficiently homogenous convenience sample of controlled studies reporting the effects of individual-level interventions on subjectively or objectively measured PA among community-dwelling middle-aged and older adults. In line with previous studies [40,41], we defined middle-aged and older adults as people aged 45 years and older. In January 2017, the reference lists of four current systematic reviews [26,27,42,43] and one meta-analysis [44] of studies of PA promoting interventions among (older) adults were searched for relevant studies. Additionally, a literature search was conducted in the electronic database MEDLINE via PubMed. The following search string was used: (((“physical activity”[Title]) AND “intervention”[Title/Abstract]) AND “trial”[Title/Abstract] AND ((“2005/01/01”[PDat]:”2017/01/04”[PDat]) AND (aged[MeSH] OR middle age[MeSH]))).

The screening of title, abstract, and full text was performed by one researcher of the EQUAL project (G.C.). Two researchers of the EQUAL project (G.C. and G.B.) performed the final selection of studies.

#### 2.2.2. Study Selection Criteria

A three-stage approach to study selection was applied. Peer-reviewed English-written journal articles on studies reporting the effects of individual-level interventions on subjectively or objectively measured PA among community-dwelling adults aged ≥45 years were eligible for inclusion in stage I. In order to focus on research based on current knowledge of determinants of PA and social inequalities in PA, only articles published after December 2004 were considered. No restrictions on country and follow-up duration were applied. All types of quantitative experimental and observational longitudinal study designs were eligible, provided that the intervention was compared with a no-intervention control condition (e.g., wait-listed, usual care). Besides participants’ age and gender, studies had to report baseline information on at least one further PROGRESS-Plus characteristic. PROGRESS-Plus, proposed by the Campbell and Cochrane Equity Methods Group, is an acronym which captures equity-relevant data items [45]. “PROGRESS” stands for Place of residence, Race/ethnicity/culture, Occupation, Gender/sex, Religion, Education, Socioeconomic status, as well as Social capital [46], and “Plus” considers further characteristics which may be associated with social disadvantage [47]. For the purpose of this study, in line with the equity-focused systematic review by Lehne and Bolte [27], “Socioeconomic status” (SES) was considered as a multidimensional concept, combining several aspects of an individual’s SEP, such as education, occupation, and income (e.g., operationalized using multidimensional SES indices or scales). We therefore treated “Income” as a distinct aspect and added it as a separate PROGRESS characteristic. Moreover, assuming that socioeconomically disadvantaged neighborhoods often have fewer opportunities for and more barriers to PA [48], “Place of residence” was defined as using area-level deprivation indices reflecting the socioeconomic conditions of an individual’s neighborhood. Just as SES, “Social capital” was considered as a multidimensional concept (e.g., operationalized using multidimensional indices). Finally, because age, marital status, as well as living situation (alone vs. with others) are associated with health inequalities and PA, they were considered as “Plus” characteristics.

To ensure comparability of the interventions, studies were excluded if they reported on workplace-based or environmental interventions, policies, or laws. Since we focused on individual-level PA interventions targeting the general population of middle-aged and older adults (i.e., potentially addressing everyone across the social spectrum of this target group), studies of interventions designed to exclusively focus on particular social groups of the middle-aged and older adult population (e.g., only one gender, only socially disadvantaged individuals, only specific ethnic minority groups) were also excluded. Furthermore, we excluded studies of interventions designed to exclusively focus on individuals who are functionally impaired, overweight/obese, or who have a specific medical condition. Also excluded were studies of interventions among nursing home residents and those that focused on participants receiving specialist exercise therapies (e.g., exercise programs for Parkinson’s disease). Finally, in line with a previous equity-specific re-analysis [30], studies with fewer than 100 participants in total were not considered.

To focus on a pool of studies being as homogeneous as possible and providing opportunities for analyzing equity-specific intervention effects, all studies meeting stage I eligibility criteria were re-assessed against more stringent criteria. Eligibility for inclusion in stage II required reporting of baseline information on at least one socioeconomic characteristic (i.e., occupation, education, income, composite SEP) as well as reporting on interventions where promoting PA was the main focus. Moreover, studies with fewer than 10 participants in one gender subgroup (and thus providing no opportunities for analyzing gender-related equity-specific intervention effects) were not considered. At stage III, due to pragmatic reasons (i.e., budget limitations, size of working group regarding collaboration procedure), a convenience sample of about ten representatives of studies meeting stage II eligibility criteria was intended to be selected.

#### 2.2.3. Search Results

The search strategy identified 1076 records (Figure 1). After removing duplicates, 966 records were screened based on their title and abstracts. Full texts of 93 articles were retrieved for in-depth review of which 53 articles were excluded, mostly because they reported studies with target populations including people <45 years. Of the remaining 30 studies (reported in 41 articles) meeting stage I eligibility criteria, 10 studies were excluded for the following reasons: promotion of PA was not the main focus of the intervention (*n* = 7); no socioeconomic indicator was reported (*n* = 2); the study population included fewer than 10 participants in one gender subgroup (*n* = 1; five men in intervention and two men in control group). The convenience sample selected at stage III comprised six author teams (first and senior author) representing eight studies. In February 2017, these author teams were invited to take part in the international collaboration. Five author teams representing seven studies agreed to collaborate. Additionally, two representatives of the AEQUIPA intervention trial PROMOTE were included in the collaboration without additional costs (Figure 1).

### 2.3. Characteristics of the Collaboration and Development of the Joint Equity-Specific Re-Analysis Strategy

Besides the German AEQUIPA intervention trial PROMOTE, three of the studies represented in the collaboration were conducted in the UK, three in the Netherlands, and one in Belgium (Table 1) [37,38,49,50,51,52,53,54,55,56,57,58,59,60,61]. Of each study team, two researchers designated by the respective study team were invited to attend a one-day face-to-face workshop in Bremen, Germany, in November 2018 to discuss the comparability of data and availability of indicators of social inequalities, as well as to start developing the joint strategy for the equity-specific re-analysis. The strategy was finalized in May 2019 by e-mail correspondence. The collaborating researchers represent various disciplines, including (social) epidemiologists, statisticians, health psychologists, primary care and public health researchers, as well as human movement scientists.

## 3. Results: The Joint Equity-Specific Re-Analysis Strategy

### 3.1. Definition of Exposure and Outcome Measures

For each intervention study, a dichotomous variable will be used to compare the intervention and control groups. For studies with more than one intervention group, intervention groups will be combined, and a dichotomous variable will be created indicating any intervention versus no intervention. This procedure is recommended by the Cochrane Statistical Methods Group for including studies in a meta-analysis (see Section 3.5) [62]. The primary outcome will be weekly minutes of moderate-to-vigorous physical activity (MVPA) at follow-up (T1) because this outcome can be defined in a similar manner across the collaborating studies (Table 1). As recommended by guidelines [63], activities burning ≥3 metabolic equivalents (METs) will be defined as MVPA. For objective PA measures, the standard Freedson cut-point of ≥1952 counts per minute [64], equivalent to ≥3 METs, will be used for defining MVPA. The post-intervention follow-up time closest to intervention end point will be used assuming that power is greatest at short-term follow-up due to lower rates of loss to follow-up and greatest intervention effects. If both objective and subjective measures are available in a study, then objectively collected data will be used. If only a subjective measure is available, a variable combining different domains of PA (e.g., leisure-time, transport-related, household-related, work-related PA) will be used, depending on the questionnaire used (Table 1). For studies with objective measures, sensitivity analyses will be conducted using weekly minutes of MVPA in ≥10-min bouts.

### 3.2. Choice and Definition of Indicators of Social Inequalities

Educational level as a measure of SEP and gender will be considered as indicators of social inequalities because both (1) are available in all collaborating studies (Table 1), (2) can be defined in a similar manner across different countries, and 3) are likely to moderate intervention effects as suggested by previous research [25,26,27,65,66,67]. According to the concept of ‘individual agency’ [15], individual-level behavior change interventions usually require notable cognitive, psychological, time, and material resources from the individual in order to be effective. As these resources tend to be socioeconomically patterned, whereby higher-SEP individuals have relatively greater resources than those with a lower SEP, interventions may be more beneficial for higher-SEP individuals [15,17,18]. This hypothesis is supported by results of systematic reviews on, inter alia, tobacco control interventions [65], obesity prevention interventions [66], and interventions to promote healthy eating [67]. Therefore, we expect higher-educated individuals to benefit more from the interventions.

Educational level will be defined according to the International Standard Classification of Education (ISCED) 2011 [68]. Based on the highest level of educational qualification or age at leaving full-time education, in each study, individuals will be grouped into three categories: “Low” (less than primary, primary, and lower secondary education or ≤16 years), “Medium” (upper secondary and post-secondary non-tertiary education or 17–18 years), and “High” (tertiary education or ≥19 years).

The decision to consider gender (only assessed as female/male) was based on the state-of-the-art theoretical foundations as well as scientific evidence regarding the impact of gender as social construct and processes on health [69,70]. According to Krieger [69] (p. 653), “gender refers to a social construct regarding culture-bound conventions, roles, and behaviors for, as well as relations between and among, women and men (…).” Previous studies indicate gender differences in the domains in which people prefer to engage in PA as well as in motivating factors and context preferences for PA (e.g., PA format, location, social setting) [23,24]. Thus, it seems plausible that interventions may also be differentially effective among males and females. This hypothesis is consistent with the results of two equity-focused systematic reviews and a pilot review in the area of PA promotion indicating that some interventions may be more effective in men than women and others vice versa [25,26,27].

### 3.3. Statistical Analyses

#### 3.3.1. Social Inequalities in Adherence and Dropout

The majority of studies included in the collaboration provides information on social indicators for study participants, but not for non-participants. Therefore, it is not possible to investigate social inequalities in intervention “reach” as this requires calculating response rates by social group [71,72]. However, census data will be consulted to compare the study population with the target population of each study, considering the studies specific eligibility criteria.

SEP and gender inequalities in intervention adherence and dropout will be assessed given that both may lead to social inequalities in intervention benefit [16,73]. Depending on the characteristics of the intervention (Table 1) and availability of data, adherence will be defined as, amongst others, use of intervention materials, attendance at group meetings, and completion of PA diaries. Dropouts will be defined as individuals with valid information on PA at T0 but without valid information at T1. Descriptive statistics (e.g., means and standard deviations, percentages) will be used to describe adherence and dropout by gender and education.

#### 3.3.2. Equity-Specific Intervention Effects

The main intervention effect will be assessed for each study, defined as the difference between the intervention and control groups in minutes of MVPA per week at T1. To do so, post-intervention values of minutes of MVPA per week will be regressed on intervention versus control group, PA value at baseline, age in years, gender, and education. Where necessary, analyses will be adjusted for cluster effects. To investigate equity-specific intervention effects, interaction terms between the grouping variable and the social indicator will be added to the main model. The *p*-values for the interaction terms as well as effect estimates with corresponding 95% CI for males, females, low-, medium-, and high-educated individuals will be calculated and reported. The included studies were originally not designed for analyzing equity-specific intervention effects and are therefore likely to lack statistical power to detect interaction effects. Therefore, both *p*-values for the interaction terms and social group-specific effect estimates with 95% CI will inform the interpretation of potential social inequalities in intervention efficacy. Moreover, absolute and relative measures of inequalities will be considered because expressing effects of inequalities in absolute or relative terms can lead to divergent conclusions regarding the impact of interventions on health inequalities [74].

Analyses will be conducted by intention-to-treat, analyzing individuals according to the group to which they were initially assigned, whether or not they adhered to the intervention. Individuals without valid information on age, gender, education, as well as PA at T0 and T1 will be excluded from the analyses (i.e., complete case analysis). Sensitivity analyses will be conducted to assess the impact of missing values. For example, multiple imputation (MI) methods for imputing outcome data for individuals without valid information on PA at T1 will be applied. For the variables age, gender, and education, MI methods will only be used if the proportion of missing values is >10%.

#### 3.3.3. Secondary Analyses

To examine the impact of the choice of social indicator and the length of follow-up time, two secondary analyses will be conducted. First, SEP is considered a multidimensional construct, comprising not only education, but also other socioeconomic indicators, which are likely to operate through different causal pathways and may differ in relevance by age and gender [75,76,77]. Thus, investigating differential intervention effects across other SEP measures may result in different findings and conclusions regarding the existence and extent of social inequalities within intervention benefits. This issue will be addressed by investigating social differences in intervention adherence, dropout, and efficacy using income/area-level deprivation as indicator of social inequalities (available for five studies, Table 1). To do so, tertiles will be calculated and individuals will be grouped into three categories: “Low” (lowest tertile), “Medium” (second tertile), and “High” (highest tertile). Moreover, due to its association with health inequalities and PA [78,79], the re-analyses will also be conducted for marital status (available for all studies). In each study, individuals will be grouped into the categories “having a partner” or “not having a partner” (including single, separated, or divorced individuals). Second, potential changes in equity-specific intervention effects over time will be investigated using weekly minutes of MVPA at a later follow-up assessment (T2) as the outcome (available for five studies).

### 3.4. Risk of Bias Assessment of Included Studies

The methodological quality of each study will be assessed using the revised Cochrane risk-of-bias tool for randomized trials (RoB 2.0) [80] and the ROBINS-I tool (Risk Of Bias In Non-randomized Studies—of Interventions) [81]. The assessment will be conducted by two researchers of the collaboration independently (one from the contributing study, the other from the EQUAL project team). Any disagreements will be resolved through discussion.

### 3.5. Data Synthesis

Results of each study concerning equity-specific intervention adherence, dropout, and efficacy will be presented in a narrative synthesis in conjunction with tabular and graphical illustrations. If feasible, intervention effect estimates of individual studies will be pooled using random-effects meta-analysis. To investigate equity-specific intervention effects, following the approach used by Love et al. [33], meta-regressions on the social indicators of interest will be performed in a meta-analysis model pooling the subgroups from each trial for these indicators. Moreover, a pooled effect estimate with corresponding 95% CI will be calculated and presented for each subgroup of interest, while tests of heterogeneity of effect between subgroups will be assessed with meta-regression [82]. The I^2^ statistics will be calculated to assess the level of heterogeneity. Sensitivity analyses will be performed to investigate possible sources of heterogeneity (e.g., study quality and study design).

Should meta-analysis be deemed inappropriate, alternative approaches will be used to synthesize und visualize the results. For example, the harvest plot, proposed by Ogilvie and colleagues [83], has been shown to be useful for synthesizing evidence on equity-specific intervention effects from heterogeneous studies, allowing demonstration of the direction of equity-specific effects in relation to the studies’ methodological quality.

### 3.6. Timeline

Studies were selected in January 2017. Authors of the selected studies were consulted for the first time in February 2017 during project proposal, and again in March 2018 after launch of the second funding phase of the project EQUAL. A first face-to-face workshop to start working on the common analysis strategy was conducted in November 2018. The strategy was finalized in May 2019 by e-mail correspondence, and the application of the strategy by the individual study teams started in July 2019. In August 2019, a Skype meeting was held to discuss first insights relating to the application of the strategy and resolve potential difficulties. A second face-to-face workshop will be held in October 2019 for discussion of the results of the analyses, reasons for potentially observed equity-specific effects, and implications for the future development of interventions. The findings will be disseminated through conference presentations and a publication in an academic peer-reviewed journal.

## 4. Discussion

Interventions to promote PA may be differentially effective across different population subgroups and thus may unintentionally increase social inequalities in PA and PA-related health outcomes. However, the potential for assessing equity-specific effects of PA interventions has not yet been exploited. Re-analyzing existing intervention studies by indicators of social inequalities could provide insight into the impact of interventions on social inequalities in PA. As this requires access to and analyses of primary data, a collaborative approach involving authors and researchers of the primary studies is needed. To our knowledge, this international interdisciplinary collaboration is the first to jointly develop and apply a strategy for systematically re-analyzing the effects of existing intervention studies on social inequalities in PA among community-dwelling middle-aged and older adults. Besides inequalities in intervention efficacy, the developed strategy will be used to investigate social inequalities in intervention adherence and dropout, allowing for a comprehensive assessment of social inequalities in intervention benefit [16,73].

A particular strength of this study is the collaborative approach involving researchers from various relevant disciplines, including (social) epidemiologists, statisticians, health psychologists, primary care and public health researchers, as well as human movement scientists. Going beyond other projects on equity-specific re-analyses [29,32,39], the jointly developed re-analysis strategy includes harmonizing the definitions of exposure and outcome measures, the choice and definition of indicators of social inequalities, as well as modeling strategies across studies as much as possible. The collaboration procedure, comprising regular exchange via e-mail, Skype, and face-to-face meetings, further bears the advantages of discussing methodological issues, analysis findings, and implications for the future development of social inequalities-sensitive interventions.

There are certain limitations to this study. The studies included in the collaboration were initially not designed for investigating equity-specific intervention effects and thus may have limited power for detecting differential intervention effects across social groups. Therefore, the equity-specific re-analysis of each study will be considered as exploratory and results have to be interpreted with caution. If feasible, individual studies will be pooled using meta-analysis, which will increase statistical power and thus improve statistical precision and credibility of social inequalities-related findings. We also acknowledge that the establishment of the collaboration to jointly develop and test the application of the equity-specific re-analysis strategy was not based on a comprehensive search for relevant studies. The sample of studies included in the collaboration were not identified via a systematic review of the literature and therefore cannot be considered to be representative of the field. Moreover, due to pragmatic reasons (i.e., budget limitations, size of working group regarding collaboration procedure), only a convenience sample of seven out of 20 eligible studies was included in the collaboration. Because we will test the application of the developed re-analysis strategy among only a convenience sample of relevant studies, the generalizability of our re-analysis results may be limited (e.g., only studies from north-western European countries are included). In this respect, it should be noted that the primary aim of this study is to develop and test the application of a strategy for re-analyzing equity-specific intervention effects as part of an international interdisciplinary collaboration, without claiming to provide an exhaustive summary of the current evidence or a definite strategy. We plan to disseminate the strategy among studies beyond our collaboration to further expand the evidence regarding the effects of interventions on social inequalities in PA among middle-aged and older adults. Finally, inclusion was restricted to a homogeneous pool of studies of individual-level PA promoting interventions. We therefore encourage future studies to re-analyze other types of interventions regarding their effects on social inequalities in PA, such as contextual-level interventions aimed at modifying the social or built environment.

## 5. Conclusions

The application of the proposed equity-specific re-analysis strategy within and beyond our collaboration will help to extend the limited evidence regarding the effects of interventions on social inequalities in PA among middle-aged and older adults. Information on the social distribution of intervention effects is a prerequisite for the design and implementation of interventions not further increasing the health gap between different social groups or, even better, reducing health inequalities. The findings of this study will be of interest to policy makers, researchers, and practitioners in the area of PA promotion targeting middle-aged and older adults.

## Figures and Tables

**Figure 1 ijerph-16-03195-f001:**
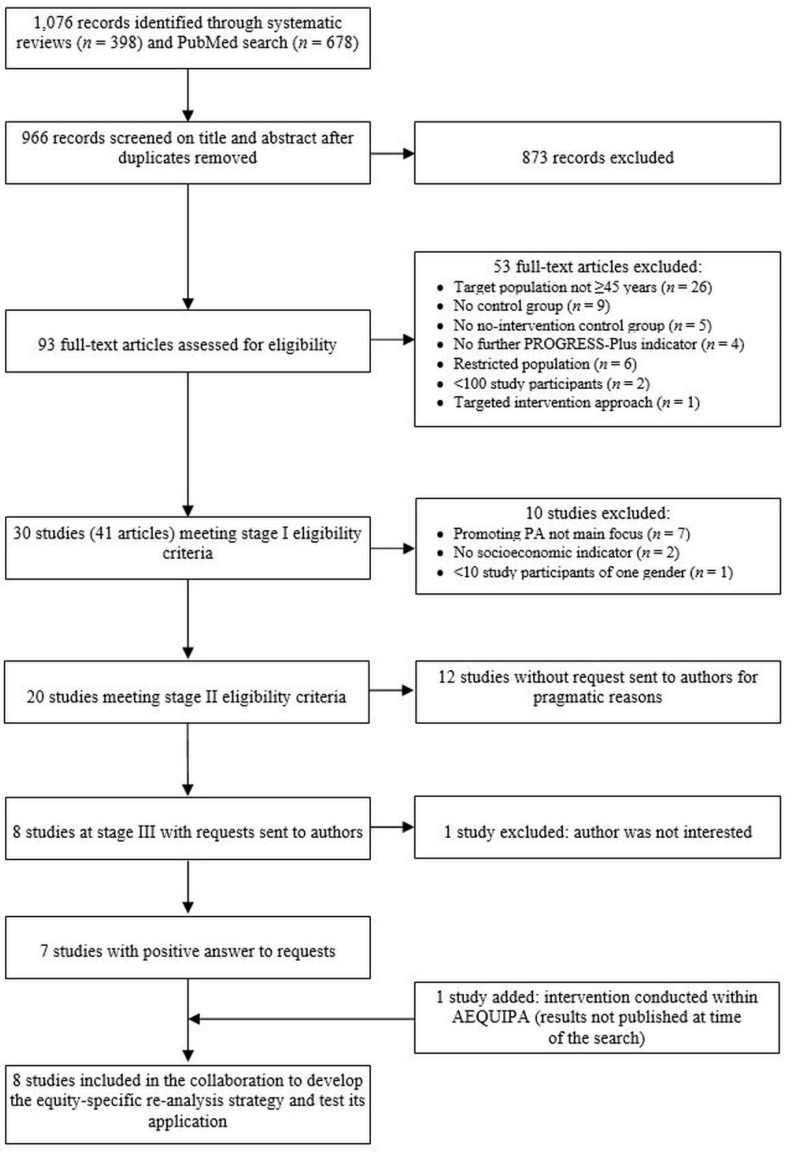
Flow chart of study selection.

**Table 1 ijerph-16-03195-t001:** Characteristics of intervention studies taking part in the collaboration.

Intervention Study	Location	Study Design, *n* *	Intervention	PA Outcome	Social Indicators
Active Plus (first version) [49,50,51,52]	The Netherlands	Cluster RCT, IG1 (2 MHC) *n* = 652, IG2 (2 MHC) *n* = 733, CG (2 MHC) *n* = 586 Follow-up = 2 and 8 months after end of intervention	IG1: Three tailored letters; personalized PA advice targeting psychosocial determinants during 4 months IG2: Intervention of IG1 plus tailored environmental information CG: Wait-listed	Self-report: Dutch SQUASH (weekly minutes of total PA, transport walking and cycling, leisure walking, gardening doing odd jobs and cycling, sports)	Gender, education, age, marital status
Active Plus (revised version) [53,54,55]	The Netherlands	Cluster RCT, IG1 (1 MHC) *n* = 439, IG2 (2 MHC) n = 423, IG3 (1 MHC) *n* = 435, IG4 (1 MHC) *n* = 432, CG (1 MHC) *n* = 411 Follow-up = 2 and 8 months after end of intervention	IG1: Three tailored letters; personalized PA advice targeting psychosocial determinants during 4 months IG2: Web-based version of intervention of IG1 IG3: Intervention of IG1 plus tailored environmental information IG4: Web-based version of intervention of IG3 CG: Wait-listed	Self-report: Dutch SQUASH (weekly days and minutes of total PA, transport walking and cycling, leisure walking, gardening doing odd jobs and cycling, sports)	Occupation, gender, education, income, age, marital status
Every step counts! [56]	Belgium	Controlled before and after study, IG (32 meeting points) *n* = 469, CG (12 meeting points) *n* = 154 Follow-up = end of intervention	IG: 10-week pedometer-defined walks in weekly walking schedules (fitness tailored and structured in walking load) CG: Wait-listed	Self-report: adapted version of GLTEQ (scores for low-, moderate-, and vigorous-intensity PA, total PA score)	Gender, education, social capital **, age, marital status
GALM [57,58]	The Netherlands	Cluster-randomized trial, IG (6 neighborhoods) *n* = 163, CG (6 neighborhoods) *n* = 152 Follow-up = end of intervention	IG: Weekly sessions emphasizing tailored moderate-intensity recreational sports activities over 15 weeks CG: Wait-listed	Self-report: Voorrips PA questionnaire, compendium of physical activities by Ainsworth et al. (energy expenditure for recreational sports activities, gardening, doing odd jobs, transport walking and cycling)	Gender, education, age, marital status, living situation
PACE-Lift [59]	UK	Cluster RCT, IG (118 households) *n* = 150, CG (117 households) *n* = 148 Follow-up = end of intervention, 9 and 45 months after end of intervention	IG: Four tailored primary care nurse-delivered PA consultations over 3 months, pedometer and accelerometer feedback, individual PA diary and plan CG: Usual care	Objective: Accelerometer (average daily step-count, weekly minutes of MVPA) Self-report: GPPAQ (being inactive, moderately inactive, moderately active), short IPAQ (time in MVPA weekly, time spend walking weekly)	Area-level deprivation, race/ethnicity, occupation, gender, education, social capital **, age, marital status, living situation
PACE-UP [60]	UK	Cluster RCT, IG1 (307 households) *n* = 339, IG2 (310 households) *n* = 346, CG (305 households) *n* = 338 Follow-up = end of intervention, 9 and 33 months after end of intervention	IG1: Pedometers, patient handbook, PA diary including individual walking plan over 3 months IG2: Intervention of IG1 plus 3 tailored practice nurse PA consultations CG: Usual care	Objective: Accelerometer (average daily step-count, weekly minutes of MVPA) Self-report: GPPAQ (being inactive, moderately inactive, moderately active), short IPAQ (time in MVPA weekly, time spend walking weekly)	Area-level deprivation, race/ethnicity, occupation, gender, education, social capital **, age, marital status, living situation
ProAct65+ [61]	UK	Cluster RCT, IG1 (14 practices) *n* = 410, IG2 (14 practices) *n* = 387, CG (14 practices) *n* = 457 Follow-up = end of intervention, 6, 12, 18 and 24 months after end of intervention	IG1: Home-based exercise program over 6 months comprising exercises, walking plan, visits of trained peer mentors IG2: Community-based exercise program over 6 months comprising instructor-delivered group exercise class, home exercise, advice to walk CG: Usual care	Self-report: CHAMPS, Phone-FITT, PASE (weekly minutes and days of MVPA)	Area-level deprivation, race/ethnicity, occupation, gender, education, income, age, marital status, living situation
PROMOTE [37,38]	Germany	RCT, IG1 *n* = 211, IG2 *n* = 198, CG *n* = 180 Follow-up = end of intervention	IG1: Tailored exercise plan; website with PA diary, online-forum, social features; weekly group meetings over 10 weeks IG2: Intervention of IG1 plus PA tracker CG: Wait-listed	Objective: Accelerometer (e.g., average daily step-count, weekly minutes of MVPA) Self-report: IPAQ	Race/ethnicity, occupation, gender, education, income, social capital **, age, marital status, living situation

Abbreviations: IG = intervention group; CG = control group; PA = Physical activity; MHC = Municipal Health Councils; Dutch SQUASH = Dutch Short Questionnaire to Assess Health Enhancing Physical Activity; GLTEQ = Godin Leisure-Time Exercise Questionnaire; IPAQ = International Physical Activity Questionnaire; MVPA= Moderate-to-Vigorous Physical Activity; CHAMPS = Community Health Activities Model Program for Seniors; PASE = Physical Activity Scale for the Elderly.* The numbers reported for sample size (n) correspond to the numbers of individuals who completed the baseline questionnaire and were assigned to the intervention or control group. ** Social capital is considered a generic term covering various operationalizations.

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
