# Peer review of "Equity-Specific Effects of Interventions to Promote Physical Activity among Middle-Aged and Older Adults: Development of a Collaborative Equity-Specific Re-Analysis Strategy"

_ijerph, 2019, doi:10.3390/ijerph16173195_

Round 1

Reviewer 1 Report

Overall comments

Thanks for the opportunity to review this paper. The overall aims of this work are laudable and a guide on how to run equity analysis would be a useful addition to the literature, but the paper uses a number of poorly specified definitions of key concepts and inclusion criteria have been used rather flexibly. I recommend these criteria are reconsidered and better justifications for exclusions are given, or, a larger number of authors are consulted to supply data to the process and the methods section of the paper re-written with these inclusions. 

On the whole, I would also recommend the goals of this paper be clarified and the write up re-structured based on this - is it a qualitative process design paper that describes how to set up a collaboration between authors? Or a guide for others to use if they wish to conduct equity analysis? Or a study protocol explaining forthcoming work? At present - I’m not sure which. I would recommend the latter. 

In the present form, the exclusion criteria that authors have used to reach their sample of collaborators are poorly enforced and it feels more that these criteria have been applied post-hoc to justify the group that was consulted. If authors wish to continue with the current format of the paper, I think they need to consult with authors as obtain data from a further 15 papers which I don’t think they have the right justification to exclude. I note more detailed comments below. 

Abstract 

Sentence: ‘The study is carried out in the EQUAL project within the federally funded German prevention research network AEQUIPA.’ Modify the English ‘carried out as part of’ 

later in the text, please give a more detailed justification for defining middle and older age as >45. Why this cut off (not 40, or 65?) 

sentence: ‘The strategy comprises the investigation of social inequalities in intervention adherence, dropout and efficacy, allowing for a comprehensive assessment of social inequalities within intervention’ doesn’t quite make sense. 

Should it read: ‘Through this process we developed a strategy that is intended to be used to investigate social inequalities in intervention adherence, dropout and efficacy. This will allow for more comprehensive assessment of social inequalities in intervention effects’? Also see my comments above - is this the goal (to create a guide?), or are you simply writing up a protocol?

Introduction

paragraph 1: ‘These social inequalities in PA have been discussed to play an important role in explaining health inequalities’ - why? Brief explanation for how PA links to health inequalities and spell out (briefly) the various research on the social patterning of PA according to the demographic characteristics you are interested in (which groups do less of which type of PA e.g. frequency/ intensity/ type). 

Paragraph 2: add a sentence to reference the fact that there may also be intervention generated inequalities because of how the interventionist treats different population subgroups (e.g. those more similar in SEP to the interventions may get better treatment) 

Paragraph 2 -give a clear definition and an example of an intervention targeting low SEP groups and a more universal approach so that the reader can appreciate the difference. Link these back to your argument that the targeted interventions require more personal resources (which resources and why?)

Paragraph 3 - you may also wish to discuss the findings of the humphries (2013) review here as they seem relevant

Paragraph 3. Sentence: ‘Although insufficient statistical power is a valid reason, the importance of understanding how interventions affect health inequalities make re-analyzing intervention studies by indicators of social inequalities a potentially valuable approach’. Please explain this to the reader in more detail (e.g. equity analysis requires interaction or subgroup analyses to compare different groups which should be sufficiently powered if you want to believe the findings from them are credible)

Also - paragraph 3 is very long! Can you make it a little shorter or split in 2? 

This introduction should include a more detailed justification for why you are interested in older adults in relation to equity analyses? Do you suppose that these individuals are doing less PA than younger age groups, or are more at risk of inequalities in health according to other factors like gender? Who are you concerned about here ( the young middle age or very old?) and link this rationale to your exclusion criteria around age used in the paper (eg. 45-80). My thoughts are that this is a very large and heterogenous group spanning 35 years, with different needs and probably different equity concerns. 

Related to which, why did you choose to focus just on older age groups, rather than look at differential effects by age in a sample of adults aged >18? Why not compare equity effects across age strata (e.g. gender differences in 18-25 v 26-35 v 36-45 etc)? 

Method 

2.1. first sentence: see comments above on the phrasing.

2.1 What is the overall goal or aim of the AQUIPA programme? Just a sentence here to give context. You don’t need the funding info here either. 

2.2.1 why did you choose the 4 systematic reviews and 1 meta analysis that you did as the basis of your search? And - you give your search strategy but don’t explain the search target here (e.g. what type of study, in which population, looking at what type of intervention and what type of PA?). This is in the next section and should be moved up. 

Related - How are you sure that this list of 5 reviews covers all reviews that could be relevant to your work? 

2.2.2 sentence: ‘For the purpose of this study, place of residence was defined as using area-level deprivation indices reflecting the socioeconomic conditions of an individual’s neighborhood.’ - why did you come to this conclusion? 

2.2.2 sentence: ‘Income was added as an additional “PROGRESS” dimension and age, marital status, as well as living situation (alone vs. with others) were considered as “Plus” characteristics.’ Should income instead be included as a measure of ‘Socioeconomic status’, and marital status a measure of ‘social capital’? Why add these into ‘plus’ as a separate category?  I think you need some clearer definitions of which measures you accepted under each progress category and which are exclude, and why. 

2.2.2. If you are interested in universal rather than targeted interventions, why did you exclude ‘environmental interventions, policies, or laws’? Aren’t these great examples of universal interventions? What, then, are you defining as universal interventions if not these? 

2.2.2 sentence: ‘so were studies whose study participants, as a result of the eligibility criteria, were restricted regarding their functional or weight status, or underlying medical condition (e.g., inclusion criterion was being “functionally impaired”, “overweight”, or “having dementia”). This is a little hard to understand - check the phrasing 

Sentence: ‘At this stage, studies with the least opportunities for analyzing equity-specific intervention effects and those with the lowest levels of comparability were not considered.’ - This doesn’t sound very systematic. Can you explain in more detail what you mean here and give a better justification for the exclusions. 

2.2.3 - I don’t agree with author exclusion criteria and think that authors of a number of the excluded papers should have been included in the consultation (a total of 15 additional). 

This includes the 8 removed because of imbalance in gender, 1 removed because of age >80, 1 removed because of a post hoc subgroup analysis, 3 removed again for small sample size re: gender, 1 Chinese study removed and 1 study of recent retirees removed. For reasons, see my comments below. 

It feels like the group of authors convened for further work is done mainly on convenience (e.g. all North Western European authors) rather than review inclusion/exclusion criteria. This is fine, but the paper should state this clearly and discuss the limitations of this approach and what they may have missed by not inviting authors of excluded papers. If the goal is mainly to test out the approach of analysing equity effects then a convenience sample is fine - you just can’t generalize the findings necessarily, but it’s useful as a guide for other authors. 

2.2.3 unless the one study with mean age >80 contains participants who were in residential facilities (I.e. the sample was free living), I think you should include this paper focusing on older adults in your analysis given this is your target population.

2.2.3 the exclusions you outline in this section don’t match your exclusion criteria stated earlier on. I.e. you note no restrictions on country but take out the study conducted in a China? This study should be included unless you can give a different justification for why not. 

2.2.3 - studies you couldn’t include because the gender ratio was >70% one gender. This probably shouldn’t be a blanket exclusion criteria, but instead depends on the sample size of the 30%. Ie. A study of 5000 people with 30% male still has sufficient sized male subgroup to conduct equity analysis. Hence, I think you should include all studies that report gender differentials, unless the sample size is so small that it is likely to be vastly underpowered (contact authors and conduct a post-hoc power analysis to check this) 

2.2.3 why exclude a study of recently retired adults? This should be included unless the sample is <45 as per your stated inclusion criteria.

2.2.3 - what’s the justification for removing the study which has done a post hoc subgroup analysis? This may or may not be underpowered but should still be included as it’s still a differential effect analysis and most of the differential analyses in the existing literature are underpowered and post hoc!

Table 1: why include studies of interventions that contain tailored content if the goal is to look at universal interventions? This ties into my earlier point - what do you define as universal interventions? Active Plus, GALM and PROMOTE specifically state their interventions are tailored (a.k.a. individualised)? How, then, do you claim these as universal - what’s the definition you are working with? Also, if your focus is on studies that are looking at just older people, then these are, by definition, interventions ‘targeted’ at a single subgroup (old people) - and hence not universal. 

Results 

Im still a little unclear even at this stage what you are trying to do here - is it an analysis protocol on how to synthesise findings from multiple papers to form one equity analysis (e.g. an individual patient data meta analysis?), or is this a guide for other authors on how to analyse equity effects? I presumed the latter but the first part of the results reads as the former. 

Keep the tenses in this section all in the same format - past or present tense if written as a guide to others, future tense if written as a protocol for future work.

3.1 sentence: ‘ For studies with more than one intervention group, intervention groups will be combined and a dichotomous variable will be created to increase statistical power.’ Justify why this approach rather than comparing separate arms (original authors should have powered their study correctly for multiple arm comparisons, and subgroup analysis are usually always underpowered as they are not the randomised comparison, so im not sure your justification holds).

3.1 sentence: ‘The primary outcome will be weekly minutes of moderate-to-vigorous physical activity (MVPA) at follow-up’ - so, did you convert the outcome measures in studies that don’t report MVPA to be in the same format (e.g. did you harmonise the data)? Or, did you just exclude them? Explain here and give the method you will use for harmonisation is this is the intention.

3.2 why only mention three indicators here from the progress plus list? Did studies not measure others? There seem to be many more listed in the last column of table 1 - why aren’t these discussed in this section?

3.5 - data synthesis - you can run an ‘individual patient data meta analysis’ on the combined data from these studies if you harmonise the outcome mesures correctly and have access to the orginal data supplied by the authors. This will allow you to then run a synthesized regression analysis where you predict MVPA from group assignment, adding in interaction terms between group and your progress factors of interest (groupXgender) using data from all studies. Please look into this technique and add a description to the analysis section to explain that you will use this if deemed relevant. If you need guidance - I’m happy to send through publications to support if needed as I’ve run this technique on a different collection of RCTs for the same purpose of equity analysis. 

If not, please provide further details of how you will look at equity effects using random effects meta analysis (e.g. subgroup analysis)? Give the reader more guidance and details of the analysis steps.

Reviewer 2 Report

Overall a well-written and interesting paper.

I find it interesting that the paper is about a strategy, but does not have results. The results read more like a Methods than a Results Section.

I believe that a little more should be to explain why gender was selected as a social indicator. Some data that shows that women are are indeed at lower SES than males in these countries would be helpful. It may be necessary to say that these results only apply to a couple of countries in Northern/Western Europe.   

Round 2

Reviewer 1 Report

All fine!